# Relationships among Structure, Physicochemical Properties and In Vitro Digestibility of Starches from Ethiopian Food Barley Varieties

**DOI:** 10.3390/foods13081198

**Published:** 2024-04-15

**Authors:** Bilatu Agza Gebre, Zekun Xu, Mengting Ma, Berhane Lakew, Zhongquan Sui, Harold Corke

**Affiliations:** 1Department of Food Science & Technology, School of Agriculture and Biology, Shanghai Jiao Tong University, Shanghai 200240, China; bilatuagza@gmail.com (B.A.G.); xuzekun@sjtu.edu.cn (Z.X.); mmt9497@sjtu.edu.cn (M.M.); 2Department of Food Science & Nutrition, Ethiopian Institute of Agricultural Research, Addis Ababa P.O. Box 2003, Ethiopia; 3Ethiopian Institute of Agricultural Research, Addis Ababa P.O. Box 2003, Ethiopia; berhanekaz@yahoo.com; 4Biotechnology and Food Engineering Program, Guangdong Technion—Israel Institute of Technology, Shantou 515063, China; 5Faculty of Biotechnology and Food Engineering, Technion—Israel Institute of Technology, Haifa 3200003, Israel

**Keywords:** food barley, chain length distribution, pasting properties, thermal characteristics

## Abstract

Studying diversity in local barley varieties can help advance novel uses for the grain. Therefore, starch was isolated from nine Ethiopian food barley varieties to determine starch structural, pasting, thermal, and digestibility characteristics, as well as their inter-relationships. The amylose content in the varieties significantly varied from 24.5 to 30.3%, with a coefficient of variation of 6.1%. The chain length distributions also varied significantly, and fa, fb1, fb2, and fb3 ranged from 26.3 to 29.0, 48.0 to 49.7, 15.0 to 15.9, and 7.5 to 9.5%, respectively. Significant variations were also exhibited in absorbance peak ratios, as well as thermal, pasting, and in vitro digestibility properties, with the latter two parameters showing the greatest diversity. Higher contents of amylose and long amylopectin fractions contributed to higher gelatinization temperatures and viscosities and lower digestibility. Structural characteristics showed strong relationships with viscosity, thermal, and in vitro digestibility properties. Cross 41/98 and Dimtu varieties are more suitable in functional food formulations and for bakery products. These results might inspire further studies to suggest target-based starch modifications and new product development.

## 1. Introduction

Barley (*Hordeum vulgare* L.) is an important cereal crop that is consumed worldwide as a whole grain. The crop is cultivated across a wide range of areas globally and ranks fourth in production, following wheat, rice, and maize. As a comprehensive nutritious cereal, barley contains different chemical components such as starch (60–70%), proteins (11–14%), lipids (4–7%), and ash (1.5–2.5%) [1]. Barley consumption can reduce the risk of metabolic disorders, including type 2 diabetes, cholesterol issues, and cardiovascular diseases, due to the high levels of dietary fibers (β-glucans and arabinoxylan), vitamins, essential minerals, and phytochemicals [2]. However, barley is greatly underutilized in the food industry. Only 2% is directly consumed by humans, with approximately 98% of barley production used for animal feed and malting [3]. 

Ethiopia has a rich history of cultivating barley, and is well-known for its wide range of barley landraces. Barley is a major staple food in the Ethiopian highlands. As a key ingredient in many local beverages, it is deeply ingrained in the dietary habits of the population, and is thus used in more diverse ways than other cereals. Different breeding programs for malt and food barley varieties are proposed in Ethiopia. The malt barley varieties are specifically bred for brewing purposes, whereas the food barley varieties are cultivated for consumption as a staple food. The food barley varieties play a crucial role in traditional Ethiopian recipes like *besso*, *chiko*, and *kolo*, as well as in local beverages such as *tella*, *borde*, and *areki*. Food barley varieties are selected mainly based on their grain yield and agronomic merits. Malt barley varieties are primarily selected based on their malting qualities. Thus, it is important to investigate the diversity of the food barley varieties with a specific focus on food quality traits for unlocking the full potential of the varieties in diverse applications. 

Similar to other cereals like wheat and rice, the quality and potential applications of barley-based products are profoundly influenced by the characteristics of barley starches. As the primary constituent of the grain, starch plays a pivotal role in determining the texture, consistency, and functional properties of barley-derived foods and non-food products [1]. Barley starches have considerable variability in composition, structure, and properties due to genetic and environmental factors [4,5]. Different genotypes can vary in the enzymes responsible for starch synthesis to produce a different structure of starch molecules. Some genotypes may produce starch molecules with different branching patterns, amylose/amylopectin ratios, or crystalline structures, affecting starch properties [6].

Previous studies have highlighted the correlation between the structural characteristics and the functional properties of barely starch. Pycia et al. [7] characterized the physicochemical properties of starches isolated from nine barley varieties and reported a correlation between pasting and rheological properties. Källman et al. [8] also associated the thermal properties of barley starch with structural characteristics. They found that gelatinization temperatures were related to clusters, and retrogradation was promoted by building blocks and amylopectin short chains. Chen et al. [9] found variation in physicochemical properties, mouthfeel, and eating quality using three varieties of waxy and non-waxy barley starches. Liang et al. [10] used three barley varieties to reveal the key roles of the amylose/amylopectin ratio and crystalline structure in determining the in vitro digestibility of starch. Chen et al. [11] also examined the significant contribution of barley kernel structure and starch structural order on in vitro digestibility using five barley starches. Recent investigations on highland barley indicated that starch structure is an essential intrinsic factor determining starch digestibility. However, previous studies were conducted with limited numbers of varieties and starch quality parameters, and multidimensional relationships need to be investigated for more clear understanding. In addition, the barley varieties characterized are mainly from Asia [12,13,14,15,16,17], Europe [7,10,18], and North America [19,20,21]. Varieties from geographical areas like Africa, specifically the Nile basin where barley is believed to be first domesticated [22], are largely neglected. Therefore, the diversity of barley and the inter-relationships between traits need to be further explored by studying a greater number of landraces from different regions. This can enable target-based selection and modification of starches for specific applications. This study aimed to (1) determine structural, pasting, thermal, and in vitro digestibility characteristics of selected food barley starches, and (2) to delve the structure–property inter-relationships of food barley starch.

## 2. Materials and Methods

### 2.1. Materials

Barley varieties were provided by the National Barley Research Program of the Ethiopian Institute of Agricultural Research (Holeta, Ethiopia). Nine food barley varieties that are currently produced and used as parent materials in barley breeding were selected (Table 1). All the varieties were grown at the research site of the Holeta Agricultural Research Center under uniform agronomic conditions. One kilogram of grain was collected from each variety for starch extraction. The enzymes amyloglucosidase (EC 3.2.1.3, 300 AGU/mg) and pancreatin (EC 232.468.9, 228 USP/mg), as well as guar gum and other chemicals needed for starch extraction and amylopectin fractionation, were purchased from Sigma Aldrich (St. Louis, MO, USA). The amylose/amylopectin and D-glucose assay kits, isoamylase (1000 U/mL, EC 3.2.1.68), and pullulanase (700 U/mL, EC 3.2.1.41) were supplied by Megazyme International Ireland Ltd. (Bray, Ireland).

### 2.2. Starch Isolation

An alkaline steeping method was used to extract starch following the procedure of Kumar et al. [23] with modifications. Barley grains were soaked in 0.1 M NaOH at room temperature for 24 h, and ground with distilled water using a blender. The resulting mixture was then passed through 100- and then 200-mesh sieves, with additional distilled water added until no visible starch was released. The filtrate was then centrifuged at 3000× *g* for 15 min to remove the supernatant and top layer. The remaining white starch at the bottom was resuspended in distilled water and subjected to the centrifugation process until a pure water supernatant was obtained. Finally, the starch was dried in a hot air oven at 37 °C and ground into a powder.

### 2.3. Amylose Content

Amylose was measured with an Amylose/Amylopectin Assay Kit (Megazyme International Ireland Ltd., Bray, Ireland) following the instructions provided by the kit.

### 2.4. Chain Length Distributions

The amylopectin sample fractionated by Kong et al. [24] was debranched using isoamylase (from *Pseudomonas* sp., 1000 U/mL, EC 3.2.1.68, Megazyme International Ireland Ltd.) and pullulanase (from *Klebsiella planticola*, 700 U/mL, EC 3.2.1.41, Megazyme International Ireland Ltd.). Then, debranching was carried out following the method of Kong et al. [25], with slight modifications. To debranch the amylopectin, 9 mg was dissolved in 450 µL of 100% DMSO and stirred with a magnetic stirrer at 100 °C for 1 h. The solution was then diluted with 2.25 mL of distilled water, 300 µL of 0.5 M sodium acetate buffer (pH 4.5), and 1 µL each of isoamylase and pullulanase. The solution was kept at room temperature with constant stirring overnight. After stopping the reaction by heating, the solution was then centrifuged and filtered using a 0.45 µm pore size filter. The chain length distributions of the debranched amylopectin were analyzed using a high-performance anion-exchange chromatography (HPAEC) system (Dionex ICS-6000 SP, Thermo Fisher Scientific, Waltham, MA, USA) with a Carbo-Pac PA-100 column (250 mm × 4 mm), following the method described by Kong et al. [24].

### 2.5. X-ray Diffraction (XRD)

To determine the long-range order of the starches, an X-ray diffractometer (MiniFlex600, Rigaku, Tokyo, Japan) operating at 40 kV and 15 mA was used. The diffractograms were obtained within an angular range of 2θ from 5 to 35°. The relative crystallinity was calculated by dividing the crystalline portion by the sum of the total amorphous and crystalline portions.

### 2.6. Fourier Transform Infrared Spectroscopy (FT-IR) 

The FT-IR spectra of the starch powders were analyzed using a Nicolet 6700 FT-IR spectrometer with a Smart iTR diamond ATR accessory (Thermo Scientific, Pittsburgh, PA, USA). The starch powders were scanned in 32 consecutive scans at a resolving power of 4 cm^−1^, ranging from 4000 cm^−1^ to 550 cm^−1^ wavenumbers. The spectrum between 1200 cm^−1^ and 800 cm^−1^ was deconvoluted using the OMNIC 8.2 software. The absorbance ratios of 1041/1014 and 1014/992 were then calculated, as described by Van Soest et al. [26].

### 2.7. Pasting Properties

The pasting properties were investigated through a Rapid Visco-Analyzer (RVA 4500, Perten Instruments, Hägersten, Sweden). In an aluminum crucible, 1.96 g of starch at dry base was carefully weighed and mixed with 26.04 mL of distilled water to form a 7% starch suspension. The analysis involved the following steps: the sample was initially equilibrated at 50 °C for 1 min, then heated to 95 °C in 3.42 min, kept at 95 °C for 2.30 min, cooled to 50 °C in 3.48 min, and finally equilibrated at 50 °C for 2 min. The paddle rotation speed was initially set at 960 rpm for the first 10 s for thorough mixing, followed by a constant speed of 160 rpm for the remainder of the testing period.

### 2.8. Thermal Properties

A Differential Scanning Calorimeter (DSC 2500) (TA Instruments, New Castle, DE, USA) was used to measure the thermal characteristics of the starches. A quantity of 2.0 mg of starch was weighed in an aluminum pan. Distilled water (6 μL) was added to the pan and thoroughly mixed with the starch. The mixture was left to equilibrate at room temperature for 24 h. Subsequently, the DSC machine was used to scan the mixture within a temperature range of 30–120 °C, with a heating rate of 10 °C per min. TA software (v5.1.1) was utilized to determine the gelatinization enthalpy (ΔH), as well as the onset (T_o_), peak (T_p_), and conclusion (T_c_) temperatures of the gelatinization process [27].

### 2.9. In Vitro Starch Digestibility 

The digestion of raw and gelatinized starches was evaluated following the procedures described by Sui et al. [28]. Raw starch (550 mg, db), 0.05 g of guar gum, 10 mL of buffer, 10 mL of distilled water, and 15 glass beads were thoroughly mixed. A mixture of pancreatic supernatant (4.5 g of pancreatin from porcine pancreas in 30 mL of distilled water) and 3.9 mL of amyloglucosidase was prepared as the enzyme solution. Each sample tube received 5 mL of the enzyme solution and was incubated at 37 °C in a shaking water bath. The enzymatic reaction was stopped by mixing the hydrolysates with 20 mL of 80% ethanol, followed by centrifugation at 1810× *g* for 15 min. The amount of released glucose was quantified using the GOPOD assay kit. The contents of rapidly digestible, slowly digestible, and resistant starch were determined according to the method described by Englyst et al. [29]. To estimate the digestibility of gelatinized starch, 550 mg (db) powder samples were boiled in 10 mL of distilled water for 20 min and then followed a similar digestion procedure as the raw starch. 

### 2.10. Statistical Analysis

Statistical data analysis was conducted using the IBM-SPSS-20 software (IBM, Armonk, NY, USA). The significance was determined using an ANOVA and Duncan’s test at a confidence level of 95% (*p* ≤ 0.05). Normalized data were used for Pearson correlation and principal component analysis (PCA). The PCA was executed using Origin 2023b (Origin Lab, Northampton, MA, USA). All analyses were carried out in triplicate and the results are expressed as the mean ± SD.

## 3. Results and Discussion

### 3.1. Amylose Contents

The amylose content of barley starch can be influenced by the genotype and the environment, and can be selected to achieve specific desired properties [1,3]. The amylose content in nine tested varieties significantly varied from 24.5 to 30.3%, with a coefficient of variation (CV) of 6.1%. All were classified as normal barley starch (Table 2). Previous reports on normal barley have shown percentages ranging from 20 to 30%, with an average of 27% [3,7]. However, although all nine varieties tested are popular in Ethiopia, the level of variation in amylose content is not as wide as desired to supplement wide applications of the crop. The proportion of amylose and amylopectin influences the stability, texture, and digestibility of products containing starch [3]. In particular, to optimize the formulation of functional foods for glycemic control, it is advisable to use starches with higher amylose contents. This is because amylose, compared to amylopectin, is less digestible due to structural features that impede enzyme attachment. Barley starch granules had a reduced interior surface area with a high concentration of granular surface-bound proteins, leading to a barrier effect against α-amylase attack [30,31]. In this study, the starch from Cross 41/98 had highest amylose content, making it potentially more suitable for formulating functional foods. This finding is further supported by the low in vitro digestibility result (Table 3).

### 3.2. Amylopectin Chain Length Distributions 

The chain length distribution (CLD) of the nine food barley varieties was classified based on DP into fa (DP 6–12), fb1 (DP 13–24), fb2 (DP 25–36), and fb3 (DP *>* 36). The results of the CLD are presented in Table 2. There were significant differences in all CLD fractions, and the fa, fb1, fb2, and fb3 ranged from 26.3 to 29.0, 48.0 to 49.7, 15.0 to 15.9, and 7.5 to 9.5%, respectively. With a CV of 7.6%, the barley starches exhibited a wider variation in amylopectin long chains (DP > 36) than short and medium chains. The starch samples displayed a typical CLD for amylopectin, with the highest peak at DP 12 or 13, and they exhibited an A-type crystalline arrangement (Section 3.3). Starches with this arrangement generally contain a higher ratio of short chains to long chains [32]. Specifically, barley starches seem to have a higher proportion of short chains than other cereal starches such as maize and rice [1]. As anticipated, all varieties had the highest proportions of fb1 (over 48%) and the lowest proportions of fb3 (below 10%). Comparable trends have been reported in native starches of highland barley [10], hulless barley [16], and rice [33]. The length of starch chains can greatly impact the formation of crystalline polymorphs and the lamellar structure in polymers. Longer chains produce thicker lamellae, resulting in more organized and densely packed crystalline structures, leading to low digestibility [11,30]. Among the nine varieties, Dimtu had the highest proportion of fb3 (DP > 36), which is consistent with the RS content.

### 3.3. X-ray Diffraction (XRD)

XRD enables an understanding of the arrangement of glucose units within the crystalline structure of starch, providing valuable insights into its unique characteristics and potential behavior in food and biomaterial development. By analyzing the absorption peaks and their precise angles, one can determine the arrangement of starch molecules and differentiate different sources of starch. All nine starch samples displayed prominent XRD absorption peaks at approximately 15°, 17°, 18°, and 23° (2θ) (Figure 1). This observation confirms that the barley starches possess the typical A-type structure, which is characteristic of cereal starches [2]. 

The XRD pattern of starch exhibits sharp peaks for the crystalline regions and dispersive curves for the amorphous portions, reflecting the semi-crystalline nature of starch [14]. These specific regions were used to determine the relative crystallinity, and the range of relative crystallinity varied from 16.1% (HB 1307 and HB 1966) to 18.0% (Cross 41/98). Numerous studies have documented a wide range of relative crystallinity levels in starches obtained from highland barley. These studies have reported values ranging from 11 to 43% [14], 18 to 33% [8], and 29 to 35% [11]. These variations may be attributed to factors such as growth location, crystal size, quantification methods, and the nature of double helices [1,14,16]. The starch from Cross 41/98 exhibits a higher RC, indicating potential use in enhancing the stability and textures of food products in a variety of formulations.

### 3.4. Fourier Transform Infrared Spectroscopy (FT-IR)

The short-range ordered structure in starch refers to the formation of a double helix in the crystalline regions, which is formed by the combination of amylose and short branch chains of amylopectin [9]. By subjecting a starch sample to infrared light and measuring the resulting absorption and transmission patterns, FT-IR can provide information about the specific molecular bonds and functional groups present in the sample. These data can then be used to determine the presence and arrangement of the double helix structure in the starch. The FT-IR spectra of nine food barley starches are presented in Figure 1, while peak ratios are summarized in Table 2. The observation of certain absorbance peaks at 3285, 2928, 1078, and 993 cm^−1^ in all samples indicated the presence of hydroxyl, alkane, carbonyl, and alkene functional groups, respectively [23].

The 800–1200 cm^−1^ bands are known distinctive features of starch, as they represent the stretching vibrations of C–C, C–OH, and C–H bonds [34]. Specifically, the absorbance intensity at approximately 1045, 1022, and 995 cm^−1^ significantly changed in starch conformation. The first two bands were related to the crystalline and amorphous regions in starch, whereas the band at 955 cm^−1^ was associated with the packing of starch double helices [34,35]. In this study, three peaks were observed at 1041, 1014, and 993 cm^−1^. Consequently, the ratio of 1041/1014 cm^−1^ was employed to assess the degree of order, while the ratio of 1014/993 cm^−1^ was utilized to approximate amorphous-to-ordered carbohydrate structure [34]. The barley starches displayed comparable resonance peaks, indicating a lack of discernible difference in their FT-IR spectra. However, there were notable differences in the peak ratios observed for both 1041/1014 and 1014/993. The range for the 1041/1014 ratio was found to be between 0.55 and 0.58, while the range for the 1014/993 ratio varied from 0.85 to 0.90, suggesting that the arrangement of starch at a short distance is affected by variety. Certain arrangements of starch in the short range can affect the level of organization, thereby impacting starch functionality. These variations can be important in various applications, such as in the food industry for texture modification and in the pharmaceutical industry for controlled drug release.

### 3.5. Pasting Properties

The pasting properties are presented in Table 4. Viscosity is a crucial factor that influences the quality of starch-based products during the heating process. It is primarily determined by the intermolecular bonding and rigidity of starch granules, as well as the density of packed starch granules [4]. Except for peak time (6.43–7.00 min, CV: 3.1%), all the parameters showed wide variation, with peak, holding, and final viscosity ranges of 744–1076 (CV: 10.8%), 648–885 (CV: 8.8%), and 827–1088 (CV: 8.4%) mPa·s, respectively. The diversity in pasting characteristics due to genotypic differences is consistent with previous findings [36]. Major factors contributing to the variation in barley starch properties include granule size, starch purity, amylose/amylopectin ratio, the nature of double helices, and processing stability. In RVA curves, the starch granules undergo expansion and increase in viscosity during the initial heating phase. As a large number of granules swell, the peak viscosity is achieved. However, as the process continues, the curve starts to decline due to the breakdown of the granules (holding viscosity). When the temperature is lowered, the viscosity increases from its minimum value to reach a final viscosity [4]. It is crucial to assess these parameters in order to comprehend the behavior and qualities of starch during heat treatment, and to choose the suitable starch for specific uses. Of the nine varieties, starch HB 1965 had the highest RVA peaks, and might be more suitable as a thickening agent in various food products such as sauces, soups, and puddings [32].

The breakdown and setback viscosities varied greatly from 96 to 190 and 163 to 252 mPa·s, with CVs of 24.4 and 13.2%, respectively. Breakdown determines the disintegration of cooked starch, and generally, a lower breakdown viscosity of starch is preferred to withstand heat processing [2]. The low peak and breakdown viscosities observed in starch Belemi indicates the presence of a strong cohesive force, ensuring minimal degradation. Setback is associated with retrogradation tendency, and a lower value indicates higher stability [5]. Among the nine starches, Senef Kolo stands out as having the slowest rate of retrogradation. This particular barley variety is renowned for its use in the traditional Ethiopian snack called ‘senef kolo’. The pasting characteristics are related to the structural, thermal, and digestibility properties of starch (Table 5 and Figure 2).

### 3.6. Thermal Properties

The barley starches showed significant variations in gelatinization behavior, and the T_o_, T_p,_ and T_c_ ranged from 52.8 to 55.6, 56.2 to 61.0, and 60.0 to 64.7 °C, respectively (Table 4). Starches that have a stronger and more rigid structure require longer heating in order to undergo gelatinization. Such starches are generally best suited for applications that require stability and resistance to heat, where they can withstand high temperatures without breaking down or losing their functionality. The gelatinization temperature ranges were in agreement with the published literature on barley starches [7,8,14,16]. The difference in temperature range (ΔT: T_c_ − T_o_) also showed significant variation from 7.2 to 9.1 °C, with a CV of 7.8%. ΔT represents the melting of individual crystals and the arrangement of amylose and amylopectin in starch granules, and values indicate the strength of the starch crystals. 

Enthalpy change (ΔH) varied from 7.9 to 9.9 J/g (CV: 8.5%), with the lowest being observed for Senef Kolo, indicating that it required the least energy to melt and uncoil the crystalline structure [9]. On the other hand, the higher ΔH for HB 1966 and Cross 41/98 suggests a greater disruption of the double helix structure during gelatinization. According to previous studies, the ΔH value is generally between 6 and 11 J/g [7,9,14,16], consistent with the current study. Differences in gelatinization properties could be due to variations in their amylose content, CLD, and RC, as indicated by close inter-relationships among the parameters (Table 5 and Figure 2). 

### 3.7. In Vitro Digestibility

The in vitro digestibility of raw and gelatinized barley starches was estimated, and the rapidly digestible starch (RDS), slowly digestible starch (SDS), and resistant starch (RS) contents of the raw samples were 18.8–30.5, 58.3–66.8, and 9.9–15.9%, respectively (Table 3). A high diversity was observed in the RDS (CV: 17.0%) and RS (CV: 15.8%) content, with a relatively low diversity in SDS (CV: 4.6%) content. The largest proportions of SDS and RS, which together accounted for about 77% of the raw starch, were found in Dimitu, Cross 41/98, and Shege. Research has proven that the inherent composition of starch can hinder digestion in multiple ways, both at the molecular and physical levels. The compactness of starch in its original form and the way its chains are arranged prevent water absorption for hydration and limit the accessibility of digestive enzymes until disrupted by gelatinization [30]. 

Gelatinization significantly altered the rate of digestion, and the structural disruption and swelling of starch granules exposed a larger surface area to digestive enzymes, leading to an increase in the proportion of RDS from 22.9% to 89.9%. Recent experiments on gelatinized barley starch showed a rapid increment of digestibility in the first 20 min, and an RDS content of up to 96% is reported [23,37,38]. Furthermore, research has shown that starches with an A-type crystalline structure and a higher percentage of short chains are more readily digestible compared to B and C types [6,32], and barley starch falls into this category. Despite the substantial decrease, the range of variations in SDS (CV: 39.4%) and RS (CV: 21.4%) fractions among gelatinized starch samples was higher compared to raw starches.

### 3.8. Structure, Physicochemical, and In Vitro Digestibility Relationships

The structural (molecular, crystalline) characteristics of barley starch showed strong relationships with the viscosity, thermal, and in vitro digestibility properties (Table 5). The molecular arrangement of amylose and amylopectin determines crystalline packing, and it inter-relates with starch properties. The peak, holding, breakdown, and final viscosities exhibited a similar trend of correlations and were significantly directly correlated with amylose content, fb3, and relative crystallinity, while they were negatively correlated with fa. The RVA peaks (peak, holding, and final viscosities) increased with the amylose content as amylose contributed to form a more viscous gel compared to amylopectin. Similarly, a recent study by Zhong et al. [39] reported a direct relationship between amylose content and PV in starches of 15 rice genotypes. However, the viscosities of starch are affected by several internal and external factors, and the correlation directions reported by different authors are not always consistent [1,16,39]. Other pasting parameters such as setback and peak time showed weak relationships with starch structural parameters. 

There were strong correlations among the thermal properties and structural characteristics of starch, and higher amylose content, longer chain lengths, and higher crystallinity contributed to higher gelatinization temperatures. Specifically, T_o_, T_p_, and T_c_ were significantly positively correlated with amylose content, fb3, and relative crystallinity, but negatively correlated with fa. A higher amylose content leads to a higher gelatinization temperature, as the tightly bound amylose molecules require more energy to break their interactions and form a gel-like structure [40]. As with amylopectin chains, longer chains have more extensive intermolecular interactions and require more energy to disrupt the interactions and undergo gelatinization [30]. Higher crystallinity also leads to a higher gelatinization temperature, as the ordered structure of crystalline regions hinders the penetration of water and the gelatinization process. The ΔT exhibited a strong positive correlation solely with RC, whereas the ΔH displayed non-significant correlations with all the parameters examined. 

In vitro digestibility of raw starch was more strongly correlated with the amylose/amylopectin ratio, amylopectin chains, and crystallinity characteristics than gelatinized starch. The proportion of SDS and RS increased with increasing amylose content, relative crystallinity, and amylopectin long chains (DP > 36), but decreased with fa. For example, the highest proportion of RS and amylose content was recorded for Cross 41/98 starch, which had lowest proportion of short-chain-length fractions. Other studies using barley starches also indicated the inter-relationships among amylose content, amylopectin chains, and in vitro digestibility [10,11]. Amylose polymers have more intra-molecular hydrogen bonds and a smaller surface area, limiting α-amylase accessibility, while more amylopectin branch points increase the surface area for starch hydrolyzing enzymes. The length of amylopectin chains is also important in the crystalline arrangements, influencing the lamellar structure, and longer chains decrease the surface area of contact with the enzyme [40]. The ratios of the FT-IR absorbance peaks, R1041/1014 and R1014/993, had no significant correlation with other starch properties.

### 3.9. Principal Component Analysis (PCA)

In order to better visualize and comprehend the relationship among barley varieties and their starch properties, a PCA analysis was conducted (Figure 2). The eigenvalues of the first six principal components were higher than 1, and PC1 and PC2 accounted for 71.1% of the variability among the samples. With 57.4% variability and an eigenvalue of 14.36, PC1 explained most of the variance, which was mainly associated with amylose content, crystallinity, in vitro digestibility, RVA peak values, gelatinization temperatures (T_o_, T_p_, T_c_), fa, and fb3. PC2, on the other hand, was related to fb1, fb2, ΔH, and setback viscosity. The positions of the samples in the biplot chart revealed differences among the varieties. Notably, varieties like Belemi, HB 1307, and Senef Kolo exhibited clear separation, while EH 1493, Shege, and Dimtu were more similar to each other. 

The biplot also visualized the structure–property inter-relationships of the barley starches. Each parameter is denoted by a vector, where the length of the vector corresponds to the importance of the parameter, and the direction indicates the nature of the relationship. Parameters such as fa and RDS were positioned in the opposite direction to all physicochemical properties present in the PCA plot, implying indirect relationships. Starch properties that displayed direct significant relationships were clustered together in the same direction: gelatinization properties were found in the upper right quadrant along with the amylose content, and the right lower quadrant was occupied by pasting properties, fb3, and relative crystallinity. RDS was grouped with fa and R1014/993 in the negative zone of PC1, whereas SDS and RS were placed with amylose content, fb3, and relative crystallinity on the opposite side.

## 4. Conclusions

This study examined the structural, physicochemical, and in vitro digestibility of starch samples isolated from nine Ethiopian food barley varieties. The varieties displayed significant differences in all the measured parameters, indicating potential for utilization in a diverse range of applications. For example, starch from the Cross 41/98 and Dimtu varieties had higher amylose contents, amylopectin long chains, RC, and RS, and could be preferred to formulate functional foods aimed at blood glucose control. In addition, their resistance to heat can best suit applications that require thermal stability. Of all the parameters determined, the greatest variation was found in viscosity and in vitro digestibility properties. The differences in starch structure and properties observed can be attributed to genetic variation. Future studies will investigate the impact of genotype–environment interactions on the stability of these starch traits. However, understanding that the materials tested are popular varieties in the country, the level of diversity identified seems insufficient for a full development of diverse food/non-food applications of barley. For instance, the amylose contents (24.5–30.3%) were all in the normal category, and breeding programs should focus more on the development of high-amylose and waxy varieties. 

Barley is an important staple food in global regions; the development of varieties with food-quality traits remains limited. In the present study, we provided fundamental insights into the starch qualities of food barley and inter-relationships between structure and functional properties. This study provides direct evidence for food scientists and plant breeders in developing new food barely varieties based on their applications. 

## Figures and Tables

**Figure 1 foods-13-01198-f001:**
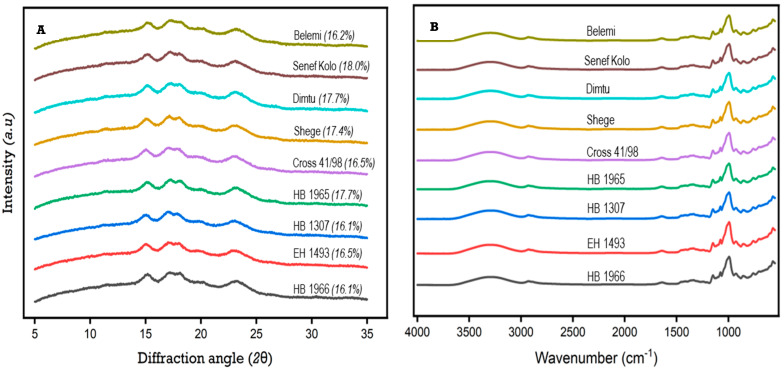
X-ray diffractograms (**A**) and FT-IR spectra (**B**) of starches isolated from Ethiopian food barley varieties.

**Figure 2 foods-13-01198-f002:**
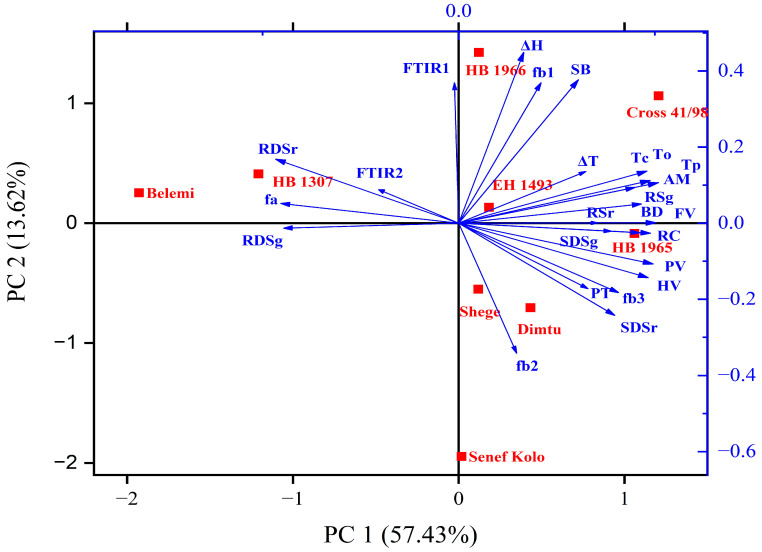
Principal component analysis biplot of food barley starches. AM: amylose; RC: relative crystallinity; FTIR1 = R1041/1014; FTIR2 = R1014/993; ΔT = T_c_ − T_o_; PV = peak viscosity; HV = holding viscosity; BD = breakdown; FV = final viscosity; SB = setback; PT = peak time; ΔT = T_c_ − T_o_; ΔH = gelatinization enthalpy; RDSr, SDSr, RSr: rapidly digestible, slowly digestible, and resistant raw starch; RDSg, SDSg, RSg: rapidly digestible, slowly digestible, and resistant gelatinized starch.

**Table 1 foods-13-01198-t001:** Information of food barley varieties.

Variety	Type	Grain Color	Row Type	Year of Release	Grain Yield at Research Field (tons/ha)
HB 1966	Released Variety	White	6	2017	3.5–5.4
EH 1493	Released Variety	White	6	2012	2.5–5.6
HB 1307	Released Variety	White	6	2006	4.8
HB 1965	Released Variety	White	6	2017	3.0–5.0
Cross 41/98	Released Variety	White	6	2012	2.5–5.6
Shege	Released Variety	White	6	1995	3.2–5.5
Dimtu	Released Variety	Purple	Irregular	2001	2.0–4.0
Senef Kolo	Farmer variety	White	6	-	-
Belemi	Farmer variety	Dark Gray	2	-	-

**Table 2 foods-13-01198-t002:** Amylose, FT-IR ratios, and CLD of food barley varieties.

Variety	Amylose (%)	FT-IR Ratios		Chain Length Distribution (CLD) (%)			
		1041/1014	1014/993	fa (DP: 6–12)	fb1 (DP: 13–24)	fb2 (DP: 25–36)	fb3 (DP > 36)
HB 1966	27.6 ± 0.55 ^cd^	0.58 ± 0.02 ^a^	0.85 ± 0.01 ^e^	27.2 ± 0.27 ^bc^	49.3 ± 0.30 ^ab^	15.0 ± 0.02 ^b^	8.5 ± 0.43 ^bc^
EH 1493	27.6 ± 0.51 ^cd^	0.55 ± 0.01 ^c^	0.86 ± 0.01 ^de^	26.7 ± 0.48 ^c^	49.7 ± 0.49 ^a^	15.0 ± 0.52 ^b^	8.6 ± 0.49 ^bc^
HB 1307	24.5 ± 0.50 ^f^	0.57 ± 0.01 ^ab^	0.87 ± 0.01 ^bc^	29.0 ± 0.15 ^a^	48.5 ± 0.16 ^de^	15.0 ± 0.04 ^b^	7.5 ± 0.04 ^d^
HB 1965	29.1 ± 0.32 ^b^	0.55 ± 0.01 ^c^	0.87 ± 0.01 ^cd^	26.4 ± 0.77 ^c^	49.4 ± 0.29 ^ab^	15.3 ± 0.19 ^b^	8.9 ± 0.54 ^ab^
Cross 41/98	30.3 ± 0.61 ^a^	0.58 ± 0.01 ^a^	0.89 ± 0.01 ^ab^	26.3 ± 0.26 ^c^	49.3 ± 0.10 ^ab^	15.5 ± 0.40 ^ab^	8.9 ± 0.53 ^ab^
Shege	28.3 ± 0.38 ^bc^	0.57 ± 0.01 ^ab^	0.87 ± 0.01 ^bc^	26.5 ± 0.49 ^c^	48.8 ± 0.21 ^bc^	15.9 ± 0.21 ^a^	8.7 ± 0.42 ^bc^
Dimtu	28.0 ± 0.10 ^bc^	0.56 ± 0.02 ^bc^	0.86 ± 0.00 ^cd^	26.6 ± 0.52 ^c^	48.7 ± 0.62 ^cd^	15.1 ± 0.12 ^b^	9.5 ± 0.21 ^a^
Senef Kolo	26.7 ± 0.79 ^de^	0.56 ± 0.01 ^c^	0.86 ± 0.01 ^cd^	27.3 ± 1.03 ^bc^	48.0 ± 0.01 ^e^	15.9 ± 0.50 ^a^	8.8 ± 0.53 ^ab^
Belemi	26.3 ± 0.57 ^e^	0.56 ± 0.01 ^bc^	0.90 ± 0.01 ^a^	28.0 ± 0.05 ^b^	48.8 ± 0.15 ^bc^	15.1 ± 0.09 ^b^	8.0 ± 0.03 ^cd^
Min	24.0	0.54	0.85	25.5	48.0	14.5	7.5
Max	31.0	0.60	0.91	29.1	50.2	16.4	9.7
Mean	27.6	0.57	0.87	27.1	48.9	15.3	8.6
SD	1.67	0.013	0.015	0.96	0.56	0.43	0.65
CV (%)	6.1	2.3	1.7	3.5	1.2	2.8	7.6

Means with the same letter in a column are not significantly different (*p* > 0.05); FT-IR: Fourier transform infrared, DP: degree of polymerization, CV = (SD/Mean) × 100%.

**Table 3 foods-13-01198-t003:** In vitro digestibility of raw and gelatinized food barley starches.

Variety	Raw Starch			Gelatinized Starch		
	RDS (%)	SDS (%)	RS (%)	RDS (%)	SDS (%)	RS (%)
HB 1966	24.7 ± 0.74 ^c^	60.3 ± 1.00 ^cd^	15.1 ± 1.04 ^a^	89.2 ± 0.70 ^d^	4.0 ± 0.64 ^ab^	6.9 ± 0.57 ^b^
EH 1493	21.7 ± 1.54 ^d^	62.8 ± 2.29 ^bc^	15.6 ± 0.92 ^a^	89.5 ± 0.80 ^cd^	3.5 ± 0.46 ^bc^	7.0 ± 0.81 ^b^
HB 1307	27.7 ± 0.86 ^b^	62.4 ± 0.70 ^bc^	9.9 ± 1.19 ^c^	92.6 ± 0.87 ^ab^	1.9 ± 0.57 ^d^	5.6 ± 0.55 ^cd^
HB 1965	21.7 ± 1.00 ^d^	66.0 ± 1.01 ^a^	12.3 ± 0.12 ^b^	86.5 ± 0.85 ^e^	5.0 ± 0.74 ^a^	8.6 ± 0.30 ^a^
Cross 41/98	19.7 ± 0.57 ^de^	64.5 ± 0.82 ^ab^	15.9 ± 0.70 ^a^	86.8 ± 1.19^e^	5.1 ± 1.20 ^a^	8.1 ± 0.57 ^a^
Shege	20.1 ± 1.32 ^de^	64.3 ± 2.42 ^ab^	15.6 ± 1.10 ^a^	92.1 ± 1.00 ^ab^	2.7 ± 0.57 ^cd^	5.3 ± 0.52 ^d^
Dimtu	18.8 ± 1.50 ^e^	66.8 ± 1.30 ^a^	14.3 ± 0.64 ^a^	91.0 ± 1.02 ^bc^	2.5 ± 0.55 ^cd^	6.5 ± 0.59 ^bc^
Senef Kolo	21.7 ± 1.39 ^d^	64.1 ± 2.06 ^ab^	14.2 ± 0.81 ^a^	88.1 ± 1.40 ^de^	5.0 ± 0.85 ^a^	6.9 ± 0.55 ^b^
Belemi	30.5 ± 0.70 ^a^	58.3 ± 0.52 ^d^	11.2 ± 0.91 ^bc^	93.7 ± 1.11 ^a^	2.1 ± 0.59 ^d^	4.2 ± 0.70 ^e^
Min	17.1	57.7	9.1	85.4	1.4	3.4
Max	31.2	68.1	16.6	94.7	6.3	8.9
Mean	22.9	63.3	13.8	89.9	3.5	6.6
SD	3.9	2.9	2.18	2.62	1.38	1.41
CV (%)	17.0	4.6	15.8	2.9	39.4	21.4

Means with the same letter in a column are not significantly different (*p* > 0.05); RDS = rapidly digestible starch, SDS = slowly digestible starch, RS = resistant starch, CV = (SD/Mean) × 100%.

**Table 4 foods-13-01198-t004:** Pasting and thermal properties of food barley starches.

Variety	Pasting Properties						Thermal Properties				
	PV (mPa·s)	HV (mPa·s)	BD (mPa·s)	FV (mPa·s)	SB (mPa·s)	PT (min)	T_o_ (°C)	T_p_ (°C)	T_c_ (°C)	ΔT (°C)	ΔH (J/g)
HB 1966	943 ± 16.9 ^c^	772 ± 14.2 ^e^	171 ± 3.1 ^b^	997 ± 20.3 ^b^	225 ± 6.1 ^b^	6.63 ± 0.06 ^b^	55.5 ± 0.12 ^a^	59.7 ± 0.44 ^b^	63.4 ± 0.40 ^cd^	7.9 ± 0.31 ^ef^	9.9 ± 0.31 ^a^
EH 1493	936 ± 20.2 ^c^	798 ± 17.9 ^d^	138 ± 3.5 ^c^	1008 ± 20.0 ^b^	210 ± 3.0 ^d^	6.67 ± 0.15 ^b^	55.3 ± 0.26 ^ab^	59.6 ± 0.21 ^b^	63.4 ± 0.46 ^cd^	8.1 ± 0.21 ^de^	8.3 ± 0.30 ^de^
HB 1307	807 ± 5.0 ^e^	716 ± 2.5 ^f^	90 ±6.7 ^d^	909 ± 6.0 ^d^	192 ± 7.6 ^f^	6.87 ± 0.06 ^a^	53.9 ± 0.15 ^c^	58.0 ± 0.36 ^c^	62.2 ± 0.44 ^e^	8.3 ± 0.29 ^cd^	9.1 ± 0.31 ^bc^
HB 1965	1076 ± 22.0 ^a^	885 ± 17.2 ^a^	190 ± 5.8 ^a^	1088 ± 23.0 ^a^	203 ± 6.9 ^e^	6.90 ± 0.10 ^a^	55.9 ± 0.67 ^a^	60.4 ± 0.71 ^a^	64.4 ± 0.56 ^ab^	8.5 ± 0.12 ^bc^	9.4 ± 0.36 ^ab^
Cross 41/98	1008 ± 7.2 ^b^	821 ± 4.6 ^c^	187 ± 3.6 ^a^	1073 ± 3.6 ^a^	252 ± 8.2 ^a^	7.00 ± 0.20 ^a^	55.6 ± 0.42 ^a^	61.0 ± 0.15 ^a^	64.7 ± 0.49 ^a^	9.1 ± 0.10 ^a^	9.8 ± 0.59 ^a^
Shege	897 ± 21.5 ^d^	763 ± 12.1 ^e^	134 ± 9.5 ^c^	944 ± 23.0 ^c^	181 ± 11.1 ^g^	7.00 ± 0.10 ^a^	54.9 ± 0.25 ^b^	59.5 ± 0.42 ^b^	63.8 ± 0.20 ^bc^	8.9 ± 0.21 ^ab^	8.8 ± 0.53 ^bc^
Dimtu	989 ± 8.5 ^b^	845 ± 4.5 ^b^	144 ± 4.4 ^c^	1064 ± 6.8 ^a^	219 ± 2.6 ^c^	6.80 ± 0.10 ^b^	55.5 ± 0.12 ^a^	59.6 ± 0.10 ^b^	63.1 ± 0.17 ^d^	7.6 ± 0.25 ^f^	8.2 ± 0.36 ^de^
Senef Kolo	998 ± 18.6 ^b^	821 ± 12.1 ^c^	177 ± 6.6 ^b^	984 ± 19.6 ^b^	163 ± 7.5 ^h^	6.93 ± 0.06 ^a^	54.2 ± 0.06 ^c^	58.5 ± 0.10 ^c^	61.9 ± 0.21 ^e^	7.7 ± 0.20 ^f^	7.9 ± 0.12 ^e^
Belemi	744 ± 10.5 ^f^	648 ± 4.5 ^g^	96 ± 6.0 ^d^	827 ± 8.2 ^e^	179 ± 3.8 ^g^	6.43 ± 0.06 ^c^	52.8 ± 0.20 ^d^	56.2 ± 0.30 ^d^	60.0 ± 0.06 ^f^	7.2 ± 0.15 ^g^	8.5 ± 0.10 ^cd^
Min.	734	644	86	820	156	6.40	52.6	55.9	59.9	7.0	7.8
Max.	1098	901	197	1112	259	7.20	56.6	61.2	65.3	9.2	10.5
Mean	933	786	147.5	988	203	6.80	54.9	59.2	63.0	8.1	8.9
SD	100.7	69.0	36.0	82.7	26.8	0.21	1.01	1.41	1.43	0.63	0.76
CV (%)	10.8	8.8	24.4	8.4	13.2	3.1	1.8	2.4	2.3	7.8	8.5

Means with the same letter in a column are not significantly different (*p* > 0.05); PV: peak viscosity, HV: holding viscosity, BD: breakdown, FV: final viscosity, SB: setback, PT: peak time, ΔT = T_c_ − T_o_, CV = (SD/Mean) × 100%.

**Table 5 foods-13-01198-t005:** Correlation of structural parameters with physicochemical and in vitro digestibility of barley starch.

		Molecular Structures					Crystalline Structures		
		AM	fa	fb1	fb2	fb3	RC	FTIR1	FTIR2
Molecular structures	AM	1	−0.929 **	0.548	0.308	0.746 *	0.773 *	0.099	0.095
	fa	−0.929 **	1	−0.485	−0.362	−0.872 **	−0.763 *	0.118	0.142
	fb1	0.548	−0.485	1	−0.449	0.149	0.262	−0.079	0.000
	fb2	0.308	−0.362	−0.449	1	0.317	0.333	0.106	0.064
	fb3	0.746 *	−0.872 **	0.149	0.317	1	0.743 *	−0.174	−0.290
Crystalline structures	RC	0.773 *	−0.763 *	0.262	0.333	0.743 *	1	0.138	−0.274
	FTIR1	0.099	0.118	−0.079	0.106	−0.174	0.138	1	0.070
	FTIR2	0.095	0.142	0.000	0.064	−0.290	−0.274	0.070	1
Pasting properties	PV	0.715 *	−0.759 *	0.253	0.301	0.791 *	0.824 **	−0.148	−0.426
	HV	0.669 *	−0.726 *	0.227	0.245	0.794 *	0.843 **	−0.248	−0.468
	BD	0.763 *	−0.736 *	0.273	0.375	0.693 *	0.692 *	0.063	−0.296
	FV	0.739 *	−0.748 *	0.402	0.082	0.776 *	0.884 **	−0.074	−0.393
	SB	0.614	−0.435	0.658	−0.382	0.345	0.553	0.418	−0.004
	PT	0.356	−0.336	−0.166	0.604	0.306	0.756 *	0.166	−0.158
Thermal properties	T_o_	0.744 *	−0.771 *	0.580	0.008	0.691 *	0.874 **	0.046	−0.498
	T_p_	0.802 **	−0.775 *	0.525	0.173	0.675 *	0.923 **	0.173	−0.360
	T_c_	0.760 *	−0.721 *	0.554	0.172	0.669 *	0.907 **	0.187	−0.318
	ΔT	0.551	−0.416	0.341	0.387	0.109	0.680 *	0.360	0.076
	ΔH	0.379	−0.097	0.512	−0.205	−0.152	0.308	0.630	0.099
In vitro digestibility of raw starch	RDS	−0.720 *	0.839 **	−0.160	−0.459	−0.865 **	−0.900 **	0.039	0.389
	SDS	0.501	−0.592	−0.037	0.368	0.722 *	0.867 **	−0.231	−0.306
	RS	0.694 *	−0.794 *	0.356	0.374	0.669 *	0.558	0.228	−0.326
In vitro digestibility of gelatinized starch	RDS	−0.675 *	0.614	−0.354	−0.241	−0.534	−0.635	0.040	0.237
	SDS	0.637	−0.573	0.235	0.400	0.472	0.503	0.014	−0.152
	RS	0.658	−0.602	0.445	0.072	0.542	0.720 *	−0.078	−0.311

** Correlation is significant at the 0.01 level (2-tailed); * correlation is significant at the 0.05 level (2-tailed); AM: amylose, RC: relative crystallinity, FTIR1: R1041/1014, FTIR2: R1014/993, PV = peak viscosity, HV = holding viscosity, BD = breakdown, FV = final viscosity, SB = setback, PT = peak time, ΔT = T_c_ − T_o_**_,_** ΔH = gelatinization enthalpy, RDS = rapidly digestible starch, SDS = slowly digestible starch, RS = resistant starch.

## Data Availability

The original contributions presented in the study are included in the article, further inquiries can be directed to the corresponding authors.

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
