# Peer review of "Relationships among Structure, Physicochemical Properties and In Vitro Digestibility of Starches from Ethiopian Food Barley Varieties"

_foods, 2024, doi:10.3390/foods13081198_

Round 1
Reviewer 1 Report
Comments and Suggestions for Authors
Currently, advanced methods for processing starch and starch-based technologies are being developed to better utilise the functional characteristics of this raw material. The determination of the physicochemical, structural, and digestive properties can guide the development of further applications of starch in the food industry. The authors isolated starches from nine Ethiopian food barley varieties and studied their physicochemical and functional properties. The properties of the tested starches were analysed using various research techniques.
The paper is well-written, appropriate research methods were used, correct conclusions were drawn, and the results were presented and discussed appropriately.
In order to improve the introduction, it would be beneficial to provide a comparison of the general characteristics of barley starch with starch isolated from other sources.
Additionally, in lines 76-77, the authors state „ In addition, the barley varieties characterized are mainly from Asia, Europe and North America…”. It would be appropriate to cite several papers that describe the characteristics of barley starch from these geographical areas.
The paper requires minor editorial corrections, such as checking the typeface and font size. For example, there is bold font in lines 90, 184-186.
Author Response
Response to Reviewer 1
Ref: foods-2938636, “Relationships among structure, physicochemical properties and in vitro digestibility of starches from Ethiopian food barley varieties”
Reviewer 1
Currently, advanced methods for processing starch and starch-based technologies are being developed to better utilise the functional characteristics of this raw material. The determination of the physicochemical, structural, and digestive properties can guide the development of further applications of starch in the food industry. The authors isolated starches from nine Ethiopian food barley varieties and studied their physicochemical and functional properties. The properties of the tested starches were analysed using various research techniques.
The paper is well-written, appropriate research methods were used, correct conclusions were drawn, and the results were presented and discussed appropriately.
Response: Thank you for your kind words and encouragements.
In order to improve the introduction, it would be beneficial to provide a comparison of the general characteristics of barley starch with starch isolated from other sources.
Response: We appreciated your constructive suggestion to improve the introduction. The revised version included comparisons with characteristics of barley starch and other starch resources (Line: 41-45, 51-66).
Additionally, in lines 76-77, the authors state „ In addition, the barley varieties characterized are mainly from Asia, Europe and North America…”. It would be appropriate to cite several papers that describe the characteristics of barley starch from these geographical areas.
Response: We included references from 12 studies conducted in Asia, Europe and North America on barley starch (Line: 85-87). The previous studies were aligned with this study. We highlighted in lines 70-85.
The paper requires minor editorial corrections, such as checking the typeface and font size. For example, there is bold font in lines 90, 184-186.
Response: Thank you. We have made the necessary corrections.
Reviewer 2 Report
Comments and Suggestions for Authors
The manuscript is interesting. Some improvements should be done by authors to increase overall quality.
Pg. 2, 2.1. Materials section: If available, it will be of great interest if authors can present the most recent production quantity (in tons) of each of the nine food barley varieties, in Ethiopia.
Pg. 8, lines 294-295: “Viscosity plays a crucial role in controlling the processability of starch-based products.” Authors should present some examples to better elucidate the meaning of this sentence.
Author Response
Response to Reviewer 2
Ref: foods-2938636, “Relationships among structure, physicochemical properties and in vitro digestibility of starches from Ethiopian food barley varieties”
Reviewer 2
The manuscript is interesting. Some improvements should be done by authors to increase overall quality.
Response: Thank you for your valuable comments. All the inputs are applied in the revised version of the manuscript.
Pg. 2, 2.1. Materials section: If available, it will be of great interest if authors can present the most recent production quantity (in tons) of each of the nine food barley varieties, in Ethiopia.
Response: We appreciate the suggestion. We added Table 1, which provided the information about the nine food barley varieties in Ethiopia including the production quantity.
Pg. 8, lines 294-295: “Viscosity plays a crucial role in controlling the processability of starch-based products.” Authors should present some examples to better elucidate the meaning of this sentence.
Response: Thank you. In the revised version, it is modified as “Viscosity is a crucial factor that influences the quality of starch-based products during the heating process. It is primarily determined by the intermolecular bonding and rigidity of starch granules, as well as the density of packed starch granules [4]” (Line: 285-288).
Reviewer 3 Report
Comments and Suggestions for Authors
The manuscript submitted for review, entitled: Relationships among structure, physicochemical properties and in vitro digestibility of starches from Ethiopian food barley varieties, is correct.
the title and abstract are compatible with each other.
However, I believe that the goal is too vague or sub-goals could be added.
material and methods:
this chapter is described quite well, but more detailed information is missing, e.g. what varieties were they, why were these varieties chosen, how much material was used for research?
Amylose content: literature is missing
the remaining methods are well described
Table 1, 'samples' column, are these barley names or coded samples?
the 'discussion' chapter is described correctly,
However, I believe that it could be enriched with additional literature
The literature contains only 29 items - which is not much.
Author Response
Response to Reviewer 3
Ref: foods-2938636, “Relationships among structure, physicochemical properties and in vitro digestibility of starches from Ethiopian food barley varieties”
Reviewer 3
The manuscript submitted for review, entitled: Relationships among structure, physicochemical properties and in vitro digestibility of starches from Ethiopian food barley varieties, is correct.
The title and abstract are compatible with each other.
Response: We appreciated the constructive comments given to improve the quality of our manuscript.
However, I believe that the goal is too vague or sub-goals could be added.
Response: Thank you. We revised the objective as “The study aimed to (1) determine structural, pasting, thermal and in vitro digestibility characteristics of selected food barley starches, and (2) to delve the structure-property interrelationships of food barley starch.” (Line: 91 – 93).
material and methods:
this chapter is described quite well, but more detailed information is missing, e.g. what varieties were they, why were these varieties chosen, how much material was used for research?
Response: In the revised version, we included Table 1 that provided additional information about the varieties. Furthermore, we updated the description of section 2.1 in line 96-101. Popular food barley varieties in Ethiopia that are currently under production and used as a parent material in barley breeding programs have been selected.
Amylose content: literature is missing
Response: More literatures as well as explanations are provided in the revised version (Line: 193 – 208).
The remaining methods are well described.
Table 1, 'samples' column, are these barley names or coded samples?
Response: Thank you. The names provided are the official designations of the varieties. The inclusion of Table 1 could enhance clarity in this regard. Furthermore, we updated "Sample" to "Variety" across all tables.
The 'discussion' chapter is described correctly. However, I believe that it could be enriched with additional literature.
The literature contains only 29 items - which is not much.
Response: We enhanced the discussion throughout the document according to the reviewer’s suggestion. In addition, we carefully read more related literatures and added in the reference section.